# Knowledge, attitude, and practice of antenatal exercises among pregnant women in Ethiopia: A cross-sectional study

**Balamurugan Janakiraman**[1]*, **Tsiwaye Gebreyesus**[1], **Mulualem Yihunie**[2], **Moges Gashaw Genet**[2]

**1** Department of Physiotherapy, School of Medicine, College of Health Sciences and Ayder Comprehensive Specialized Hospital, Mekelle University, Mekelle, Ethiopia, **2** Department of Physiotherapy, School of Medicine, College of Medicine and Health Sciences, University of Gondar comprehensive specialized hospital, Gondar, Ethiopia

* bala77physio@gmail.com

## Abstract

### Background

"Is pregnancy opportunity or a barrier for engaging in exercise". Maternal health still is a top priority in sub-Saharan Africa including Ethiopia. Participation in exercises during pregnancy in low-middle income countries is constrained. The objective of this study was to evaluate the knowledge, attitude, and practice of antenatal exercises among Ethiopian women during pregnancy, and also to examine the barriers to prenatal physical activity.

### Methods

A descriptive hospital-based cross-sectional study was conducted and 349 pregnant women receiving prenatal care at the ante-natal care clinic, University of Gondar comprehensive specialized hospital were recruited. Data were obtained on maternal characteristics, knowledge, attitude, practice, and barriers towards antenatal exercise (ANEx) by interview method.

### Results

Among 349 pregnant women, 138 (39.5%) and 193 (55.3%) had adequate knowledge, a positive attitude, and good practice respectively. Overall, 108 (30.9) of the respondents practiced antenatal exercise, while only 41 (37.9%) of those pregnant women had a good practice. Brisk walking (90.7%), relaxation (38.9%), and breathing exercise (36.1%) were most practice ANEx, while pelvic floor 6 (5.6%) and 3 (2.8%) yoga were the least practiced. Enhancing post-natal recovery (71%) and vaginal bleeding (64.5) were perceived as benefits and contraindication of ANEx. More than half of the pregnant women (53.6) reported that ANEx is not appropriate for Ethiopian culture. Knowledge, attitude, and practice of ANEx among pregnant women are significantly associated with higher education, government employees, pre-pregnancy exercise, and being advised on ANEx before. Women with adequate knowledge are more likely to have a good practice (AOR 4.53, 95%CI: 1.64, 15.3).

**Data Availability Statement:** All data relevant to our findings are contained within the manuscript. The full dataset contains sensitive participant information. Requests for further details on the

dataset and queries concerning data sharing should be directed to the Head of the Department, Dept of Physiotherapy, University of Gondar at getachewazeze43@gmail.com.

**Funding:** This study was fully resourced and partly funded by the University of Gondar (Grant no: SOM112/7/2019). The views presented in the article are the authors and not necessarily express the views of the funding organization. University of Gondar did not involve in the design of the study, data collection, analysis, and interpretation. There was no additional external funding received for this study.

**Competing interests:** The authors have declared that no competing interests exist.

**Abbreviations:** ACOG, American College of Obstetricians and Gynecologists; ANEx, Antenatal Exercise; ANC, Antenatal care; DM, Diabetes Mellitus; HTN, Hypertension; EC, Ethiopian calendar; EDHS, Ethiopian Demographic Health Survey; GDM, Gestational Diabetes Mellitus; KAP, Knowledge, Attitude, and Practice; NICE, National Institute for Health and Care Excellence guidelines; LBP, Low Back Pain; LMICs, Low middle-income countries; MMR, Maternal mortality ratio; SD, Standard Deviation; SPSS, Statistical Package for Social Sciences; UoGCSH, University of Gondar comprehensive specialized Hospital; PFME, Pelvic Floor Muscle Exercise; PT, Physiotherapy; UNFPA, United Nation Population Fund; UoGCSH, University of Gondar Comprehensive Specialized Hospital; WHO, World Health Organization.

## Conclusion

The findings of this study suggest that knowledge concerning antenatal exercise is low and their attitude is reasonably favorable. However, very few Ethiopian pregnant practices ANEx according to recommended guidelines during pregnancy.

## Background

Pregnancy should be seen as an opportunity to embrace exercise routines and women should be encouraged to maintain those habits. Antenatal exercises are tailored to promote health benefits to both pregnant women and fetuses [1,2]. According to the National Institute for Health and Care Excellence guidelines (NICE) and the American Congress of Obstetricians and Gynecologists (ACOG), antenatal exercise (ANEx) has minimal risks and paramount benefits, although some modification is needed as per maternal and fetal requirements. ACOG recommended that low-impact or moderate exercise for 30 minutes on most days of the week, helps with weight management, reduced risk of gestational diabetes mellitus (GDM), and improved psychological well-being [1,3].

The global maternal mortality ratio (MMR) in the year 2015 is 216, the MMR of Sub-Saharan Africa (SSA) is 546, Ethiopia is among those SSA countries with high MMR, 412, and the majority of the deaths occurred at the reproductive age group [4]. Further, for every woman who dies, an estimated 20 to 30 suffers from preventable morbidity. Maternal health is one of the top priorities for the government of Ethiopia (MOH, 2018) and other SSA countries. The United Nations in 2015 proposed 17 Sustainable Development Goals (SDGs), and UNFPA emphasized improving reproductive health care services and promotion of international maternal health standards in all the countries by 2030 [5,6]. Studies had reported that ANC that included exercise programs have an impact on the major preventable causes of fetal ill health, infant death, shape trends of mortality, and morbidity among the women population [7–9]. Over the past two decades, attitude towards antenatal exercises has changed around the world and the concept of fit pregnancy steadily gained popularity based on its positive outcome [10–12]. But still, cultural acknowledgment, ethnic practices, beliefs, maternal age, unwanted pregnancy, education level of women, health care access, healthcare utility, availability of trained women health professionals, health-seeking behavior of women, family support, and economic status are significantly associated with the knowledge and practice of antenatal exercise among women living in developing countries [8,10,12–16]. Common misconception and concerns of pregnant women, family member, and some of the obstetrician-gynecologists are that exercise during pregnancy may cause miscarriage, poor fetal growth, musculoskeletal pain, musculoskeletal injury, and premature delivery, while in the absence of absolute contra-indications, antenatal exercises are safe, desirable, vital, and should be encouraged [1,16,17].

A sedentary lifestyle before or during pregnancy is frequently associated with negative maternal health impact and poor neonatal outcomes [18]. Studies report that most of the pregnant women living in low-middle income countries (LMICs) are not sufficiently active and did not meet the present exercise guidelines during their pregnancy. The prevalence of sedentary activity is high (76–79%) among Ethiopian pregnant women and the majority of them are below par with the ACOG recommendations [19–21]. Despite physical activity and antenatal exercises are not threatening according to the current guidelines, this population's behavior does not seem to change [17].

Previous regional studies have investigated the level of physical activity, ANC service utilization in pregnancy, the timing of ANEx, antenatal depression, and gestational weight gain [22,23]. Most of them reported that many Ethiopian pregnant women are keen on exercise participation but don't practice regular physical exercise, which makes the question about the influence and belief of antenatal exercise on mother and fetal health more important [8,24,25]. Despite current evidence, pregnant women attending ANC programs in Ethiopia are often misinformed about antenatal exercises. In Ethiopia, there is scarce research on "what is known, believed, and practiced concerning ANEx. Hence a thorough understanding of current knowledge about ANEx, insight into their attitude regarding ANEx, and why some women fail to exercise in the context of the socio-economic, cultural, and educational background of Ethiopian pregnant women is imperative in building comprehensive maternal health interventions to cover all facets of risk. This study aimed to evaluate the knowledge, attitudes, and practice of antenatal exercise and association with the socio-demographic and maternal characteristics among Ethiopian pregnant women attending ANC clinic at University of Gondar comprehensive specialized hospital, Ethiopia.

## Methods

### Study design, setting, and population

An institutional-based cross-sectional study was conducted from March 30, 2019, to April 30, 2019, in Gondar city, Ethiopia. Gondar city is located around 750 km north of the Ethiopian capital, Addis Ababa, at an elevation of 2,706 meters above sea level, in the region of Amhara. The ante-natal care (ANC) clinic at a tertiary facility called University of Gondar comprehensive hospital (UoGCSH) provides health care free of cost for eight zones in the Amhara region and serves as a major referral center that has served more than 5 million people who live in North Gondar zone and neighboring zones.

North Gondar-zone has a total population of 3,654,920 of whom 1,807,289 are females and the majority of the inhabitants depend on the free healthcare provided by UoGCSH [26]. According to the Ethiopian Demographic and Health Survey (EDHS, 2016), the country-wide fertility rate, women's literacy, and employment status of women were 4.6, 42, and 33% respectively. However, in the Amhara region, this report revealed a lower fertility rate of 3.7%, a slightly higher literacy rate of 45%, and lower employment status of 27% of women aged between 15 and 49 years. About 29.8% of Ethiopian women aged between 15–49 years live in the Amhara region and the gender parity index (GPI) for the net attendance ratio (NAR)at secondary school level is highest (NAR: 1.39) in Amhara. The proportion of women aged 15–49 who received ANC in Ethiopia in 2016 has increased to 62% from 34% in 2011 but only 17% attended post-natal care in 2016. Besides, in Amhara, the health facility delivery rate is 27.1% with the same proportion (26.4%) gave birth in public health facilities, preceded by at-home 71.4%. Furthermore, only 27.7% of births were delivered by the community health workers,19.6% of these births were attended by nurse or midwife, and the majority by traditional birth attendants (55.8%) [27,28].

All pregnant women aged between 16 and 49 years attending antenatal care at the ANC clinic, UoGCSH during the data collection period were considered. Study participants who were having antenatal care during the data collection period were included. Pregnant mothers who presented and/or diagnosed with medical or obstetric complications resulting in ambulatory disabilities and serious psychological conditions that could have impacted the reliability of information were excluded from this study. Written consent was taken from the participants prior to data collection. However, for four participants who were below the age of 18 years, written consent was obtained from their parents. The study was conducted after

obtaining ethical approval from the Institutional Review Board (IRB), CMHS, University of Gondar (Ref no; SOM/112/7/2019). Permissions were obtained from the regional public health institute and the authorities of the study site prior to the study. Before enrollment, the pregnant women were informed about the study, its objectives, and its importance.

## Sample size determination and sampling technique

A sample size of 384 was calculated using a single population proportion formula based on the following assumptions: considering a 95% confidence interval, 50% prevalence, and 5% precision. A 5% non-response and attrition rate were added to the estimated sample size, the finally derived sample was 403. The study subjects were selected using a systematic random sampling method. On average, about 45 to 55 pregnant women visit the ANC clinic of gynecological units in UoGCSH every day. According to the information from the registry of the ANC clinic, a total estimated number of pregnant women visiting ANC clinic during the study period would be 1350 and the $K^{th}$ (K = 3) was determined by dividing the total number of pregnant women estimated to attend by the required sample size. The first sample was between 1 and $K^{th}$ was randomly chosen, then taking every $K^{th}$ participant thereafter, where $K^{th}$ was a sampling interval based on the register list made for the day. The procedure was repeated until the estimated eligible sample size was reached with the study period. In the case of unwillingness to participate, the immediate next participant was approached.

## Data collection tools and procedures

A structured knowledge, attitude, and practice (KAP) ANEx questionnaire (S1 File) was used for data collection. The questions to be included in the instrument were adapted from previous studies conducted in Brazil, Nigeria, Zambia, American Pregnancy Association top recommended exercises (APA), and ACOG recommendations [17,29–31]. The questionnaire requested information on the socio-demographic, maternal characteristics, their knowledge concerning ANEx, need for ANEx during pregnancy, and barriers to ANEx. During adaptation practices that are very unlikely due to cultural and feasibility issues in the Ethiopian context were omitted; for instance, items like the practice of "swimming or antenatal-aqua exercises". The Amharic version of the questionnaire was pretested by a test-retest method (by observing one week between test and retest) on 5% (n = 21) of pregnant women who were not part of the study participants. The items in the questionnaire yielded a percentage that ranged from 83.6 to 97.5%, the intra-class coefficient was 0.964 (95% CI 0.92, 0.98). Using the pretested questionnaire the data was collected by trained female data collectors by individual interviews. The data collectors were trained on the questionnaire for 2 days by BJ. This questionnaire contained four domains; the first one collected socio-demographic and maternal characteristics (such as age, education, religion, residence, employment, parity, gestational period, etc.), whereas the remaining three domains inquired the level of KAP (knowledge, attitude, and practice) regarding ANEx.

Besides, the presence of awareness about ANEx was determined by asking "do you know about these antenatal exercises by listing the different types of exercises. Knowledge (notions about ANEx) domain contained 22 questions (benefits and contraindications) of exercises with three possible responses yes, no, and I don't know. Each correct answer (yes) was assigned "1 score" and wrong answer (no) or inappropriate answer (I don't know) were given "0 score". For the knowledge scale, the scores ranged from 0 to 22. Participants who answered correctly to 22 knowledge questions, scored more than, and equal to the mean value were defined to have adequate knowledge. Attitude (opinion concerning ANEx) domain consists of 12 questions (thinking or feeling towards ANEx), with two possible responses yes or no and scored 1

or 0 respectively. Those women who answered 12 questions correctly or greater than or equal to the mean score were categorized to have a favorable attitude. If a respondent reported herself to have practiced ANEx for at least three times a week for a minimum of 20 min per session (ACOG recommendations) [1] during the current pregnancy they were categorized to have adequate practice.

## Data analysis

Data were entered into Epi info version 7.0 and exported to Statistical Package for Social Sciences software version 23.0 (SPSS Inc., Chicago, USA). The completeness, clarity, and internal consistency or missing data was checked by the PI. Descriptive statistics were computed using percentage, mean, standard deviation, and frequency distribution were used to summarize the data. Chi-square test was used to test the associations between knowledge and attitude of pregnant women towards antenatal exercises and the participant's characteristics. Alpha level was set at 0.05. The binary logistic regression was used to identify the association between the outcome and predictor variables. Predictor variables, those with significance in the univariate test (cut off $p$ 0.20) were fitted into the multivariate model. A stepwise approach was carried out to examine the association with different independent variables and the value of the final step was considered. Adjusted odds ratios with 95% confidence intervals were estimated by considering $p$-value of $< 0.05$. When a clear sub-group appeared to be present, significance testing using Pearson $\chi^2$ was conducted. This study is reported as per the Strobe checklist (S2 File).

## Results

### Socio-demographic, obstetric, and health characteristics of pregnant women

Among 403 pregnant women those who were approached, 349 pregnant women responded with written consent and participated in the study. This is 86.6% of the response rate and 90.9% of the power calculated sample size. The mean of the respondents was 27.5 years with a standard deviation of ±5.86. The most common reason for non-response was not interested.

Among the pregnant women who were surveyed, majority (58.5%) were in the age group between 25–35 years. Fifty one 51 (14.6%) women were unable to read and write, while 134 (38.4%) had diploma and degree level education, 290 (83.1%) lived in urban, and most of the women 297 (85.1%) reported that their religious affiliation was Orthodox Christian. Just over half of the women (52.7%) were unemployed, 307 (88%) were married, and three-fourth of them had a family monthly income of less than 2500 Ethiopian birr (equivalent to $< 75$ USD). Three-fifth (59.3%) of the women were in their third trimester and 90 (25.8%) of the women were overweight or obese versus 16 (4.6%) underweight. Near half of the women (47.3%) were nulliparous, one in four previous deliveries took place at home, 35 (10.1%) reported miscarriage at least once, low back pain, and weight gain was diagnosed among 7.4% and 6.6% women respectively. Participant's socio-demographic, obstetrics, and health characteristics of pregnant women are presented in (Table 1).

### Awareness, knowledge, attitude, and practice of pregnant women

Majority of the women (63.6%) reported not to have participated in regular physical exercise before their pregnancy and about two-third had never been advised about antenatal exercises. Among those (n = 133) who had been advised about ANEx, the most often advised exercise was walking (86.5%) and the least was cycling. The most commonly cited source of information about ANEx was from healthcare professionals and social health workers (33.8%).

**Table 1. Socio-demographic and maternal characteristics of pregnant women attending antenatal care clinic (ANC) at University of Gondar comprehensive specialized Hospital, Gondar, Ethiopia (n = 349).**

| Variables | | n (%) |
|---|---|---|
| Age in years | <25 years | 106 (30.4) |
| (mean27.5±5.86) | 25-35years | 204 (58.5) |
| | >35 years | 39 (11.2) |
| Residence | Urban | 290 (83.1) |
| | Rural | 59 (16.9) |
| Religion | Orthodox | 297 (85.1) |
| | Protestant | 11 (3.2) |
| | Muslim | 41 (11.7) |
| Level of education | No formal education | 51 (14.6) |
| | Primary school | 49 (14) |
| | Secondary school | 115 (33) |
| | Diploma | 65 (18.6) |
| | Degree and above | 69 (19.8) |
| Family income/month | <2500ETB | 255 (73.1) |
| | 2500-4000ETB | 57 (16.3) |
| | >4000ETB | 37 (10.6) |
| Type of family | Extended family | 32 (9.2) |
| | Nuclear family | 317 (90.8) |
| Employment status | Unemployed | 184 (52.7) |
| | Governmental | 100 (28.) |
| | Private | 26 (7.4) |
| | Merchant | 39 (11.2) |
| Have you ever smoked | Past | 11 (3.2) |
| | Current | 1 (0.2) |
| | Never | 337 (96.6) |
| Alcohol during pregnancy | Yes | 57 (16.3) |
| | No | 292 (83.7) |
| Parity | No children | 165 (47.3) |
| | 1-2children | 136 (39.8) |
| | >2children | 45 (12.9) |
| Gestational period | First trimester | 41 (11.7) |
| | Second trimester | 101 (28.9) |
| | Third trimester | 207 (59.3) |
| History of miscarriage | Never | 314 (89.9) |
| | Once | 28 (8) |
| | More than once | 7 (2.1) |
| Previous mode of delivery* | Labor | 151 (85.8) |
| | Caesarean | 35 (14.2) |
| Place of delivery* | Hospital | 123 (66.4) |
| | Home | 40 (21.6) |
| | Hospital & home | 4 (2.2) |
| | PHC | 18 (9.7) |
| BMI | Underweight: <18.5 kg/m$^2$ | 16 (4.6) |
| | Normal: 18.5–24.99 kg/m$^2$ | 237 (67.9) |
| | Overweight: 25–29.99 kg/m$^2$ | 74 (21.2) |
| | Obesity:≥30 kg/m$^2$ | 16 (4.6) |

(*Continued*)

**Table 1.** (Continued)

| Variables | | n (%) |
|---|---|---|
| Medical conditions* | None | 279 (79.9) |
| | DM | 4 (1.1) |
| | GDM | 17 (4.9) |
| | LBP | 26 (7.4) |
| | Weight gain | 23 (6.6) |

ETB-Ethiopian birr, PHC-Primary health care, DM-Diabetes Mellitus, GDM-Gestational diabetes, LBP-Low back pain

*denotes multiple answers.

Additional information on the awareness of the participants and chi-square associations are attached as an S3 File.

The mean knowledge score of pregnant women (n = 349) was 9.85 ± 2.7 and ranged from 2 to 20. Among 349 pregnant women, only 138 (39.5) had adequate knowledge scores. About 71%, 63.3%, and 51.6% correctly identified that ANEx can help enhance post-natal recovery, improve stamina, and prevent excessive weight gain respectively. Similarly, 22.6%, 23.5%, 24.9, and 32.4% of women knew that ANEx is beneficial in strengthening pelvic floor muscles, abdominal muscle strength, prevents post-natal depression, and gestational DM respectively. Table 2 shows the frequency of knowledge of ANEx in the Likert type.

The mean attitude score of pregnant women was 6.73. More than half (55.3%) of the participants had a favorable or positive attitude based on the cutoff value. Majority of them (87.4%) think that ANEx will help them get back to shape after pregnancy, 253 (72.5%) responded that ANEx should be performed based on advice, and 224 (64.2) were keen to do exercises. Only 122 (35%) of pregnant women believed that ANEx will prevent pregnancy-related complications (Table 3).

Among 349 women less than a third (30.9%) reported that they were practicing ANEx during the present pregnancy; however, the practice was categorized as adequate ($\geq$ 3 times/week and $\geq$ 20 min per session), in only near 10% of women. Walking was the most commonly practiced exercise (90.7%), while pelvic floor exercise (5.6%) and yoga (2.8%) were among the least practiced ante-natal exercises. The principal barriers to ANEx reported by women those who were not engaged in exercise were being afraid that ANEx could be harmful (67.6%), lack of time (52.7), and lack of information (37.8).The frequency of practice, adequacy, and barriers for ANEx is shown in Table 4.

## Determinants of KAP

In multivariate analyses when adjusted for the other independent variables, level of education was a significant predictor of the knowledge of ANEx among pregnant mothers. Those who completed diploma and degree level were found to be 2.8 (95% CI 1.3, 7.3) times and 3.2 (95% CI 1.2, 6.5) times more likely to be knowledgeable about ANEx than those with lesser education or no education. Pregnant women employed in the government sector were one and half times more likely knowledgeable about ANEx than the unemployed, private employees and self-employed (AOR 1.5, 95% CI: 1.1, 2.7). Pregnant women those who reported not be involved in physical activity before pregnancy, those who had 1 or 2 children, and those who were never advised about ANEx were 54% (AOR 0.46, 95% CI: 0.26, 0.80), 58% (AOR 0.42, 95% CI: 0.24, 0.75), and 83% (AOR 0.17, 95% CI: 0.08, 0.30) less likely to be knowledgeable about ANEx than their counterpart.

**Table 2. Knowledge (benefit and possible contraindications) of antenatal exercise in Likert type among pregnant women attending ANC at University of Gondar comprehensive specialized Hospital (UoGCSH), Gondar, Ethiopia (n = 349).**

| ANEx Characteristics | Response n (%) | | |
|---|---|---|---|
| | Yes | I don't know | No |
| Reduces the risk of gestational diabetes | 113(32.4) | 171(49) | 65(18.6) |
| Enhances energy and staminab | 221(63.3) | 115(33) | 13(3.7) |
| Strengthens the pelvic floor musclesb | 79(22.6) | 152(43.6) | 118(33.8) |
| Reduces risk of perinatal and postnatal back pain | 161(46.1) | 173(49.3) | 15(4.3) |
| Helps to cope with delivery painb | 157(45) | 165(47.3) | 27(7.7) |
| Reduces postnatal abdominal muscle weaknessb | 82(23.5) | 124(35.5) | 143(41) |
| Prevents excessive weight gain b | 180(51.6) | 150(43) | 19(5.4) |
| Reduces risk of HTNb | 139(39.8) | 165(47.3) | 45(12.9) |
| Enhances post-natal recovery | 248(71) | 90(25.8) | 11(3.2) |
| Individualized exercises are safe and bestb | 182(52.1) | 104(29.8) | 64(18.1) |
| Reduces the risk of post-natal depression | 87(24.9) | 115(33) | 147(42.1) |
| Chest pain during pregnancy | 123(35.2) | 145(41.5) | 81(23.2) |
| Difficulty in breathing | 139(39.8) | 141(40.4) | 69(19.8) |
| Abdominal pain during pregnancy | 159(45.6) | 121(34.7) | 69(19.8) |
| Back pain during pregnancy | 225(64.5) | 54(15.4) | 70(20.1) |
| Uncontrolled Type1 DMc | 106(30.4) | 164(47) | 79(22.6) |
| Uncontrolled HTN during pregnancyc | 152(43.6) | 123(35.2) | 74(21.2) |
| Uterine contractionsc | 150(43) | 137(39.3) | 62(17.7) |
| Vaginal bleeding | 225(64.5) | 38(10.9) | 86(24.6) |
| Premature laborc | 181(51.9) | 93(26.6) | 75(21.5) |
| Dizziness during pregnancyc | 144(41.3) | 88(25.2) | 117(33.5) |
| Decreased foetus movementc | 209(59.8) | 66(18.9) | 74(21.3) |
| **Knowledge of ANEx (Summary index)** | | | |
| Adequate knowledge | **138 (39.5)** | | |
| Inadequate knowledge | **211 (60.5)** | | |

[b]Related to benefits and

[c]related to contraindication for ANEx.

Pregnant women aged between 25 and 35 had a favorable attitude (AOR 2.1; 95% CI 1.1, 4.8) as compared to those aged above. Participants who completed diploma and degree level were about one and half times more likely to have a positive attitude towards ANEx (AOR 1.4, 95% CI: 1.1, 3.1) and (AOR 1.6, 95% CI: 1.0, 2.9) respectively. Pregnant women who work in government (AOR 2.5, 95% CI: 1.2, 5.4) and private sectors (AOR 1.95, 95% CI: 1.03, 5.2) were twice more likely to have a positive attitude towards ANEx than those who were unemployed and self-employed. Those pregnant women who were never advised about ANEx and reported not to be involved in physical activity before pregnancy were 33% (AOR 0.67, 95% CI: 0.12, 0.85) and 79% (AOR 0.21, 95% CI: 0.14, 0.59) less likely to have positive attitude towards ANEx. When adjusted for all the other predictor variables in the multivariate model, pregnant women with adequate knowledge were about 4 times (AOR 3.9, 95% CI: 1.4, 6.1) more likely to have a positive attitude towards ANEx than those with inadequate knowledge.

Urban dwelling pregnant women were 3 times more likely to engage in ANEx. Pregnant women with a higher level of education were about 3 times (AOR 2.7, 95% CI: 1.1, 8.7) and 4 times (AOR 4.0, 95% CI: 2.3, 11.4) more likely to practice ANEx than lower educated and

**Table 3. Frequency of factors influencing negative attitude among the pregnant women attending ANC towards antenatal exercise, University of Gondar comprehensive specialized Hospital (UoGCSH), Ethiopia(n = 349).**

| Variable | n(%) |
|---|---|
| **Attitude towards antenatal exercises** | |
| Positive score ($\geq$ 6.73) | 193 (55.3) |
| Negative score ($<$ 6.73) | 156 (44.7) |
| Negative attitude towards antenatal exercise (n = 156) | |
| I think ANEx is essential | 198(56.7) |
| I believe ANEx suits our culture | 187(53.6) |
| I think ANEx doesn't harm my baby | 171(49) |
| I believe ANEx will help me get back to shape | 44(12.6) |
| I have a good family support | 167(47.9) |
| I have enough time to do exercise | 126(36.1) |
| I like to do exercises | 125(35.8) |
| I think ANEx will prevent complications | 227(65) |
| I believe ANEx with help rapid post-natal recovery | 151(43.3) |
| I think if ANEx is individualized it is better | 111(31.8) |
| I think ANEx will give an energetic feel | 237(67.9) |
| I think ANEx should be done based on advice | 96(27.5) |

**Table 4. Practice, adequacy of antenatal exercise and the barriers to do antenatal exercise, as self-reported by the pregnant women, UoGCSH, Ethiopia (n = 349).**

| Variables | n (%) |
|---|---|
| Practicing ANEx in the current pregnancy | |
| Yes | 108 (30.9) |
| No | 241 (69.1) |
| Type of antenatal exercises practiced (n108)* | |
| Brisk walking | 98 (90.7) |
| Relaxation exercises | 42 (38.9) |
| Breathing exercises | 39 (36.1) |
| Ankle and toe exercises | 36 (33.3) |
| Back care exercises | 13 (12) |
| Aerobics | 07 (6.5) |
| Pelvic floor exercises | 06 (5.6) |
| Yoga | 03 (2.8) |
| Is the exercise adequate | |
| Yes | 41(11.7) |
| No | 308 (88.3) |
| Barriers to practice of ANEx (n 241)* | |
| Is afraid that it may be harmful for foetus | 163 (67.6) |
| Lack of time | 127 (52.7) |
| Lack of information | 91 (37.8) |
| Family member advice not to do | 62 (25.7) |
| Feels uncomfortable | 55 (22.8) |
| Lack of family support | 34 (14.1) |
| Feels tired | 28 (11.6) |
| Doesn't like exercising | 14 (5.8) |

*Difference in the frequency of pregnant women may reflect multiple responses.

uneducated women. Those who reported a monthly family income between 2500 and 4000 ETB were found to three and half times (AOR 3.5, 95% CI: 1.9, 7.4) more likely to be practicing ANEx than those who had more and less income. Pregnant women employed in government and private sectors were more likely to practice ANEx than unemployed and self-employed women (AOR 1.4, 95% CI: 1.0, 3.3) and (AOR 2.19, 95% CI: 1.01, 5.4) respectively. Pregnant women those who were less likely to practice ANEx were women who had 1–2 children (AOR 0.34, 95% CI: 0.15, 0.8), those who reported no physical activity before pregnancy (AOR 0.49, 95% CI: 0.09, 0.88), and who had never heard about ANEx (AOR 0.40, 95% CI: 0.10, 0.54). Multivariate regression revealed a significant association between knowledge and practice of ANEx, pregnant women with adequate knowledge were 4.5 times (AOR 4.53, 95% CI: 1.64, 15.3) more likely to practice ANEx. Moreover, positive attitude was non-significantly associated with good practice (Table 5).

## Discussion

Evidence has shown that ANEx enhances the physical and psychological well-being of pregnant women. Appropriately prescribed ANEx is safe and beneficial in improving maternal health. Despite this, exercises during pregnancy are poorly practiced in many part of the world. It is imperative that pregnant women understand the components, benefits, contraindications, and precautions to effectively practice ANEx supported by good attitude.

In this study, pre-pregnancy exercise, higher level of education, prior advice about ANEx, institutional employment, and most importantly knowledge about ANEx are found to be strong predictors for the practice of ANEx. Furthermore, Christian orthodox, unemployment and parity have been identified as predictors for not engaging in ANEx. The women in this study were shown to be not adequately knowledgeable concerning the practice of ANEx their overall mean score was just below 50% and their attitude towards ANEx was favorable; however, very few actually practiced ANEx adequately. The study sample consisted of young women with the majority of them in the reproductive age group, about two-thirds of women had school level or no formal education, and mean parity was about two. These characteristics are comparable to those found in the samples described in the regional studies by Hjorth et al. [19], Gebregziabher et al. [20] and Hailemariam et al. [30], who evaluated predictors of physical activity during pregnancy and a study by Mbada et al. [31] which reported on knowledge and attitude about ANEx among Nigerian pregnant women. About 38% (n = 133) of respondents in this study had been advised about ANEx. This is much lower than the studies in Brazil (68.1%) and India (66%). Further, the knowledge level of the pregnant women in this study was lower than studies in Brazil and India [28,32]. The differences in awareness and knowledge might be due to the majority of participants in this study had lower education, lack of education and counseling about ANEx resulting in a paucity of information, cultural belief, lack of family support, and myths in Ethiopia. However, the reported knowledge level of pregnant women in this study was higher than the other African country Zambia (19%) [9]. This could be due to the factors that lower educational status of women, beliefs that exercise does not suit their cultural and strong impact of the same predictors in this study.

Most of the pregnant women in this study reported being aware of walking, aerobics, and back exercises, and relaxation exercise. However, pelvic exercise (Kegel), cycling, swimming, and breathing exercise were mostly not known as a component of ANEx. On the contrary, the APA ranked ANExs in order as Kegel, swimming, walking, cycling, aerobics, and dance [17]. Further, lack of swimming pools, lack of swimming skills, non-affordability or non-availability of bicycles may have contributed to low awareness. Surprisingly, most of the women in this study were not aware of the much important pelvic floor exercise (Kegel exercise). In regards

**Table 5. Bivariate and multivariate logistic regression analysis result of KAP towards antenatal exercise and associated factors during pregnancy among women, Northwest Ethiopia, 2019 (n = 349).**

| Variables | Knowledge | | Attitude | | Practice | |
|---|---|---|---|---|---|---|
| | COR(95%CI) | AOR (95% CI) | COR (95%CI) | AOR (95%CI) | COR (95% CI) | AOR (95%CI) |
| Age in years | | | | | | |
| < 25 years | 1 ref | - | 1 ref | 1 ref | 1 ref | - |
| 25-35years | 0.88(0.42, 1.85) | - | 3.6(1.65, 7.7)* | **2.1 (1.09, 4.8)*** | 0.75 (0.3, 1.5) | - |
| > 35 years | 0.80(0.40, 1.60) | - | 2.4(1.2,5.0)* | 1.03 (0.73, 4.1) | 1.19 (1.01, 3.4) | - |
| Residence | | | | | | |
| Urban | 3.(1.53,5.9)* | **1.46(1.01, 3.36)*** | 3.2(1.7,5.7)* | 2.06 (1.08, 4.3) | 9.2 (1.25, 18.8)* | **3.1 (1.11, 4.5)*** |
| Rural | 1 ref | 1ref | 1 ref | 1 ref | 1 ref | 1 ref |
| Religion | | | | | | |
| Orthodox | 0.78 (0.41, 1.5) | - | 0.7 (0.38, 1.44) | - | 0.31 (0.14, 0.70)* | **0.61 (0.19, 0.94)*** |
| Protestant | 2.2 (0.56, 8.8) | - | 2.9 (0.55, 15.1) | - | 1.77 (0.42, 7.3) | 0.92 (0.42, 2.2) |
| Muslim | 1 ref | - | 1 ref | - | 1 ref | 1 ref |
| Level of education | | | | | | |
| No formal education | 1ref | 1 ref | 1 ref | 1 ref | 1 ref | 1 |
| Primary school | 2.6 (1.1,6.3)* | 0.77 (0.29, 2.0) | 2.25 (1.1, 5.0)* | 0.44 (0.15, 1.28) | 1.04 (0.14, 7.7) | 1.68 (0.70, 4.03) |
| Secondary school | 1.8 (0.80, 3.9)* | 0.99 (0.36, 2.9) | 2.65 (1.33, 5.2)* | 0.63 (0.20, 1.95) | 2.33 (1.4, 11.1) | 1.49 (0.67, 3.31) |
| Diploma | 5.1 (2.2, 11.8)* | **2.8 (1.3, 7.34)*** | 2.13 (1.0, 4.5)* | **1.4 (1.1,3.14)*** | 3.94 (1.81, 19.1)* | **2.7 (1.09, 8.7)*** |
| Degree and above | 5.0 (2.17, 11.6)* | **3.2 (1.22, 6.51)**** | 3.44 (1.61, 7.3)* | **1.6 (1.01, 2.94)**** | 8.65 (1.9, 39.2)* | **4.01 (2.3, 11.4)*** |
| Family income/month | | | | | | |
| < 2500 ETB | 1 ref | 1 ref | 1 ref | 1 ref | 1 ref | 1 ref |
| 2500–4000 ETB | 2.1 (1.03, 4.15)* | 1.4 (0.99, 5.34) | 1.71 (1.2, 3.5)* | **1.52 (0.99, 4.06)*** | 20.5 (8.8, 48.1)* | **3.5 (1.9,7.41)*** |
| > 4000 ETB | 0.73 (0.39, 1.34) | 0.21 (0.09, 1.51) | 0.62 (0.3, 1.1)* | 0.74 (0.37, 1.46) | 3.1 (1.19, 7.6)* | 0.87 (0.45, 1.7) |
| Employment status | | | | | | |
| Unemployed | 1ref | 1ref | 1 | 1 ref | 1 ref | 1 ref |
| Governmental | 2.66 (1.61, 4.4)* | **1.5 (1.12, 2.73)**** | 1.95 (1.17, 3.2)* | **2.51 (1.17, 5.4)*** | 2.14 (1.1, 4.6)* | **1.44 (0.97, 3.32)*** |
| Private | 1.59 (0.68, 3.6) | 0.63 (0.23, 1.7) | 2.35 (1.0, 5.7)* | **1.95 (1.03, 5.2)*** | 4.19 (1.7, 10.3)* | **2.19 (1.01, 5.4)*** |
| Merchant | 1.22 (0.59, 2.5) | 0.64 (0.25, 1.6) | 1.09 (0.5, 2.2) | 0.9 (0.43, 2.18) | 1.01 (0.21, 4.7) | 0.71 (0.21, 1.3) |
| Parity | | | | | | |
| No children | 1 ref | 1 ref | 1 ref | 1 ref | 1 ref | 1 ref |
| 1–2 children | 0.52(0.32, 0.83)* | **0.42(0.24, 0.75)*** | 0.53(0.34, 0.85)* | 0.51(0.31, 0.83) | 0.33 (0.15, 0.71)* | **0.34(0.15, 0.8)*** |
| >2 children | 0.49(0.24, 0.99)* | 0.93(0.39, 2.2) | 0.21(0.10, 0.42)* | 0.29(0.13, 0.64) | 0.60 (0.26, 1.3) | 0.89(0.78, 1.3) |
| History of miscarriage | | | | | | |
| Never | 1 ref | - | 1 ref | 1 ref | 1 ref | - |
| Once | 0.68 (0.3, 1.5) | - | 1.06 (0.48, 2.3) | 1.35 (0.56, 3.2) | 1.67 (0.61, 4.6) | - |
| Twice and more | 0.24 (0.1, 2.1)* | - | 0.32 (0.11, 1.7)* | 0.49(0.0, 3.3) | 0.11 (0.03, 0.72) | - |
| PA before pregnancy | | | | | | |
| Yes | 1 ref | 1ref | 1 ref | 1 ref | 1 ref | 1 ref |
| No | 0.37 (0.23, 0.59)* | **0.46 (0.26, 0.80)*** | 0.70 (0.48, 1.17)* | **0.21 (0.14, 0.59)*** | 0.33 (0.17, 0.64)* | **0.49 (0.09, 0.88)*** |
| Advised about ANEx | | | | | | |
| Yes | 1 ref | 1ref | 1 ref | 1 ref | 1 ref | 1 ref |
| No | 0.14 (0.09,0.23)* | **0.17 (0.08, 0.30)*** | 0.48 (0.31, 0.75)* | **0.67 (0.12, 0.85)*** | 0.33 (0.17, 0.64)* | **0.40 (0.10, 0.54)*** |
| Knowledge level | | | | | | |
| Adequate | | | 11.5 (1.1, 22.1)* | **3.88 (1.41, 6.13)*** | 9.53 (4.1, 22.2)* | **4.53 (1.64, 15.3)*** |
| Inadequate | | | 1 ref | 1 ref | 1 ref | 1 ref |
| Attitude | | | | | | |
| Positive | | | | | 2.5 (1.1, 3.8)* | 1.5 (1.1, 4.9) |
| Negative | | | | | 1 ref | 1 ref |

*significant at < 0.05

**significant at <0.001.

to knowledge about the benefits of ANEx, most women in this study believed antenatal exercise enhances post-natal recovery, improves stamina, and prevent weight gain. Except for the later benefit, these findings were contrasted with the other studies [9,33].

A finding in the present study which was much expected like elsewhere [28,31,34] was knowledge, attitude, and practice of ANEx was significantly higher among pregnant women with a higher level of education and paid employment. Besides, the women in this study also believed that back pain and vaginal bleeding during pregnancy are contraindications for ANEx. Except for vaginal bleeding, in the absence of underlying complications, back pain during pregnancy is at best a relative contraindication and should not rule not pregnant women from engagement in exercise according to ACOG recommendations [1,35]. Nonetheless, the findings of this study revealed that the knowledge about ANEx was influenced by education level, pre-pregnancy exercise habits, and previous advice about ANEx.

About 55% of pregnant women in this study demonstrated a positive attitude towards exercise during pregnancy. Therefore, a large group of respondents seems to have a negative attitude towards ANEx in pregnancy. This finding is below par with the recent studies that have reported a positive prototype shift in attitudes toward exercise among pregnant women globally over the past two decades [10,31,36,37]. This study found that attitude towards exercise in pregnancy was mostly influenced by concerns about fetal safety, cultural constraints, lack of family support, and insufficient information about ANEx. This finding is thoroughly contrasting to the findings elsewhere, the most reported reasons for not exercising were tiredness, uncomfortable, lack of time, and lack of motivation [28,31,38]. This unique reporting could be due to feeling of inadequacy, need for support, long working days, prioritizing family routines, and women in Africa rather feel that pregnancy is mostly a barrier to engage in exercise. And most of the participants in this study during the interview reported that pregnancy is a barrier for physical activity. Hence, pregnancy is perceived as a barrier that prevents this population from exercising despite being aware of the possible benefits of ANEx. A qualitative design study in this population is needed to further illuminate the views of non-exercising Ethiopian pregnant women.

This study found that about 30% of the pregnant women practiced exercise during pregnancy, nonetheless, only 11.7% among the overall sample and 37.9% among those who practiced exercise had adequate practice in accordance with the minimum recommended guidelines for this group [1,29,35]. This is almost similar to the findings of two regional studies reporting about level of physical activity during pregnancy [20,30]. However, this is much lower than the practice of exercise reported in Nigeria (84.7%), Canada (29%), and Brazil (29%) in a similar population [28,31,33]. This difference might be due to the knowledge level, awareness level, educational level, socio-economic differences, and more importantly the limitation in the utility of care including lack of health care counseling concerning ante-natal exercises. Also, the cut off for adequacy (ACOG recommendation; $\geq$ 3 days/week, $\geq$ 20 min/day) of practice set in this study is similar to the Canadian and Brazilian studies. The most frequent reported barriers to practice exercise in this study were risk to fetus, lack of time, and inadequate information or training, which is consistent with the barriers reported by pregnant mothers in Australia, Canada, and Brazil [28,36,37].

Further, the most reported exercise practiced by the respondents in this study was walking, breathing, and relaxation exercises. Hence, improved guidelines and counseling concerning ANEx that includes exercises that they would like to perform and exercises that are most important (pelvic floor exercise) could have a better impact on maternal wellbeing in this population.

## Limitations and strengths

To the best of our knowledge, few studies elsewhere and none regionally in the literature have evaluated the knowledge, attitude, and practice of antenatal exercises and the explanations why this population does not exercise, which possibly could have limited the comparing or contrasting the findings of this study. Further, the operational definition for adequate knowledge involves the perception of right and wrong which depends on the individual's level of access to different means of communication and life experience. Also, the close ended answers opted for knowledge on the ANEx may not explain the extent of the knowledge. Nevertheless, this interpretation was based on the recommendation of the ACOG and the population included was homogenous which could have minimized any interpretation bias. Despite these limitations, this study included a power calculated sample and we believe that the findings of this study may collaborate towards enhancing prenatal care guidance and counseling concerning antenatal exercises among healthcare professionals and policymakers in Ethiopian women's health.

## Conclusions

Our findings suggest that pregnant women's knowledge concerning the practice of antenatal exercise is comparably low to global standards and their attitude seems somewhat better. However, relatively few practiced exercises and very few exercised with adequacy recommended by the American College of Obstetricians and Gynecologists (ACOG) guidelines.

## Supporting information

**S1 File. Structured questionnaire in English version.**
(DOCX)

**S2 File. STROBE (STrengthening the Reporting of OBservational studies in Epidemiology) checklist for KAP of antenatal exercise among pregnant women; a hospital based cross-sectional study.**
(DOCX)

**S3 File. Additional tables for frequency distribution of awareness of antenatal exercise, Chi-square association KAP and pregnant women characteristics.**
(DOCX)

## Acknowledgments

### Declarations

Our gratitude and appreciation go to the data collectors, pregnant women participants, physicians, authorities of ANC clinic, University of Gondar Comprehensive Specialized Hospital (UoGCSH), and data collectors.

## Author Contributions

**Conceptualization:** Balamurugan Janakiraman, Tsiwaye Gebreyesus, Moges Gashaw Genet.

**Data curation:** Balamurugan Janakiraman, Mulualem Yihunie, Moges Gashaw Genet.

**Formal analysis:** Balamurugan Janakiraman, Moges Gashaw Genet.

**Funding acquisition:** Balamurugan Janakiraman.

**Methodology:** Moges Gashaw Genet.

**Resources:** Moges Gashaw Genet.

**Supervision:** Balamurugan Janakiraman, Moges Gashaw Genet.

**Writing – original draft:** Moges Gashaw Genet.

**Writing – review & editing:** Balamurugan Janakiraman, Tsiwaye Gebreyesus, Mulualem Yihunie, Moges Gashaw Genet.

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
