## [Decision Letter · Decision Letter 0]

31 Dec 2020

PONE-D-20-30016

Knowledge, attitude, and practice of antenatal exercises among pregnant women in Ethiopia: a cross-sectional study

PLOS ONE

Dear Dr. Janakiraman,

Thank you for submitting your manuscript to PLOS ONE. After careful consideration, we feel that it has merit but does not fully meet PLOS ONE’s publication criteria as it currently stands. Therefore, we invite you to submit a revised version of the manuscript that addresses the points raised during the review process.

An expert in the field handled your manuscript, and we are appreciative of their time and contributions. Please address ALL of the reviewer's comments in your revised manuscript.

We look forward to receiving your revised manuscript.

Kind regards,

Frank T. Spradley

Academic Editor

PLOS ONE

2. Please include additional information regarding the survey or questionnaire used in the study and ensure that you have provided sufficient details that others could replicate the analyses. For instance, if you developed a questionnaire as part of this study and it is not under a copyright more restrictive than CC-BY, please include a copy, in both the original language as well as the English version already provided, as Supporting Information.

'This study was fully resourced and partly funded by the University of Gondar. The views presented in the article are the authors and not necessarily express the views of the funding organization. University of Gondar did not involve in the design of the study, data collection, analysis, and interpretation'

a. Please provide an amended statement that declares *all* the funding or sources of support (whether external or internal to your organization) received during this study, as detailed online in our guide for authors at http://journals.plos.org/plosone/s/submit-now

Please also include the statement “There was no additional external funding received for this study.” in your updated Funding Statement.

Reviewers' comments:

Reviewer's Responses to Questions

**Comments to the Author**

1. Is the manuscript technically sound, and do the data support the conclusions?

Reviewer #1: Yes

2. Has the statistical analysis been performed appropriately and rigorously? 

Reviewer #1: Yes

3. Have the authors made all data underlying the findings in their manuscript fully available?

Reviewer #1: Yes

4. Is the manuscript presented in an intelligible fashion and written in standard English?

Reviewer #1: Yes

5. Review Comments to the Author

Reviewer #1: The authors evaluated the knowledge, attitudes, and practice of antenatal exercise and association with the socio-demographic and maternal characteristics among Ethiopian pregnant women. This is an important, but neglected subject in maternal health, particularly in African context. The findings of the study are worthwhile to inform policy direction and action in crafting intervention to address the gaps pertaining to the knowledge, attitudes and practice of prenatal physical activity.

I have few minor comments as follows:

Abstract

1. The objective of this study was to evaluate the knowledge, attitude, and practice of antenatal exercises among Ethiopian women during pregnancy, and also to find out why some women do not exercise during pregnancy” I suggest change the phrase ‘to find out why”… “to examine the barriers to prenatal physical activity”.

2. “ANC” Write in full

3. …”among them had a good practice” Revise this phrase. I suggest changing among them to pregnant women.

4. “More than half of them (53.6) think that ANEx doesn’t suit Ethiopian culture” . Again see my earlier comment in 3 above concerning them. Furthermore, write formally. Change ‘doesn’t’. The word ‘suit’ is not talking well to the context.

Methodology

5. “Pregnant mothers aged between 16 and 49 years, during any trimester of pregnancy those who were…” There is something missing in the sentence.

Results

6. “The education tally showed..” Rephrase this phrase

7. “Walking is the most accomplished” The word accomplished is not appropriate in the context.

8. “The foremost barriers to do ANEx reported by women not engaged in exercise were being afraid that ANEx could be…” This sentence need editing.

9. “In multivariate analyses when adjusted for the other independent variables, better education was a significant predictor of the knowledge of ANEx among pregnant mother”. What is better education?

6. PLOS authors have the option to publish the peer review history of their article (what does this mean?). If published, this will include your full peer review and any attached files.

Reviewer #1: No

---

## [Decision Letter · Decision Letter 1]

9 Feb 2021

Knowledge, attitude, and practice of antenatal exercises among pregnant women in Ethiopia: a cross-sectional study

PONE-D-20-30016R1

Dear Dr. Janakiraman,

We’re pleased to inform you that your manuscript has been judged scientifically suitable for publication and will be formally accepted for publication once it meets all outstanding technical requirements.

Kind regards,

Frank T. Spradley

Academic Editor

PLOS ONE

Reviewers' comments:

Reviewer's Responses to Questions

**Comments to the Author**

1. If the authors have adequately addressed your comments raised in a previous round of review and you feel that this manuscript is now acceptable for publication, you may indicate that here to bypass the “Comments to the Author” section, enter your conflict of interest statement in the “Confidential to Editor” section, and submit your "Accept" recommendation.

Reviewer #1: All comments have been addressed

2. Is the manuscript technically sound, and do the data support the conclusions?

Reviewer #1: Yes

3. Has the statistical analysis been performed appropriately and rigorously? 

Reviewer #1: Yes

4. Have the authors made all data underlying the findings in their manuscript fully available?

Reviewer #1: Yes

5. Is the manuscript presented in an intelligible fashion and written in standard English?

Reviewer #1: Yes

6. Review Comments to the Author

Reviewer #1: Thank you in providing relevant responses to the comments and issues raised in the manuscript.

I wish you good luck in your research work.

7. PLOS authors have the option to publish the peer review history of their article (what does this mean?). If published, this will include your full peer review and any attached files.

Reviewer #1: **Yes: **Prof Daniel Ter Goon

---

## [Editor Report · Acceptance letter]

11 Feb 2021

PONE-D-20-30016R1 

Knowledge, attitude, and practice of antenatal exercises among pregnant women in Ethiopia: a cross-sectional study 

Dear Dr. Janakiraman:

I'm pleased to inform you that your manuscript has been deemed suitable for publication in PLOS ONE. Congratulations! Your manuscript is now with our production department. 

Kind regards, 

on behalf of

Dr. Frank T. Spradley 

Academic Editor

PLOS ONE